# Development of management indicators of nursing for minimizing physical restraints focused on older adult patients hospitalized in acute care settings: A Delphi consensus study

**Maya Minamizaki**[1]*, **Mana Doi**[2], **Yuka Kanoya**[1]

**1** Nursing Course, School of Medicine, Yokohama City University, Yokohama, Kanagawa, Japan, **2** Chiba Faculty of Nursing, Tokyo Healthcare University, Funabashi, Chiba, Japan

* mkan@yokohama-cu.ac.jp

**Data Availability Statement:** Relevant data are within the paper and its Supporting Information files. We did not have participants' permits to share

## Abstract

Nursing management activities are important in influencing staff nurses' action to prevent or withdraw physical restraints. However, limited studies have been conducted empirically to determine the nursing management activities required for minimizing physical restraints. Therefore, there is a need for basic standards of nursing management activities to minimize physical restraints in acute care settings. This study aimed to develop nursing management indicators to minimize physical restraint (MaIN-PR) in hospitalized older adult patients in an acute care setting. It was conducted between June and October 2021 in Japan using a Delphi consensus approach. Fifty nurses working at top or middle management levels or as certified nurse specialists in gerontological nursing enrolled as participants. The potential indicators obtained from the literature review and interviews were organized inductively to develop two types of draft indicators: (1) 35 items for top management and (2) 33 items for middle management. We asked the nursing managers and certified nurse specialists in gerontological nursing to assess the validity of each indicator in three rounds. Of the 50 initial panelists, 12 from top management and 13 from middle management continued till the third round. MaIN-PR contained 35 indicators for top management and 28 indicators for middle management and were classified into the following six metrics: planning, motivating, training, commanding, organizing, and controlling. To the best of our knowledge, the current MaIN-PR are the first set of nursing management indicators for minimizing physical restraint, including perspectives on geriatric nursing in acute care settings. These indicators could guide both top and middle nursing management, thus supporting staff nurses' judgment in minimizing physical restraints to enhance the quality of older adult patient care.

## Introduction

Physical restraint constitutes any action or procedure that prevents a person's free body movement to a position of choice and/or normal access to his/her body by the use of any method,

data generated and/or analyzed publicly. Thus, the datasets during the current study are not publicly available. Data are available on request from the Institutional Review Board of the Medical Department of the Yokohama City University (rinri@yokohama-cu.ac.jp) for researchers who meet the criteria for access to data.

**Funding:** MM was supported by JSPS KAKENHI Grant Number JP21K11085 (https://kaken.nii.ac.jp/ja/grant/KAKENHI-PROJECT-21K11085/). The funders had no role in study design, data collection and analysis, decision to publish, or preparation of the manuscript.

**Competing interests:** The authors have declared that no competing interests exist.

attached or adjacent to a person's body that he/she cannot control or remove easily [1]. While these devices are used worldwide, care dependency, common among older adults, has the strongest association with physical restraint use in acute care [2]. Physical restraint causes serious physical and emotional problems [3–5] and negative outcomes, such as poor quality of life, increased fall risk, mortality, and longer hospital stays [6–9]. Thus, nurses should avoid physical restraints as the advocates to prevent further functional decline and protect the independence and dignity of older adult patients in acute care settings. However, nurses' overestimation of the safety of physical restraints complicates their decision to use or avoid physical restraints for older adult patients [10]. In addition, nurses' intention to use restraints is related to their perceived behavioral control, attitude [11], and knowledge [12]. Therefore, we believe that any strategy designed to prevent or withdraw physical restraints would need to support a change in staff nurses' knowledge, perception, and attitude toward physical restraints.

The staff nurses support strategies to minimize physical restraints include staff education on geriatric care and fall prevention [13], expert team intervention by geriatric nurse practitioners and geriatric specialist nurses [13, 14], analysis of issues from nursing records about physical restraints, and implementing relevant hospital policies to minimize physical restraints [15]. These are nursing management activities that support a change in staff nurses' knowledge and perception of physical restraints and promote geriatric nursing care. However, evidence-based findings regarding effective nursing management activities aimed at minimizing physical restraints are scarce.

In this study, we aimed to develop evidence-based indicators for nursing management to minimize physical restraints in acute care settings. We defined "nursing management indicators" as perspectives for judging and evaluating the state of nursing management activities and "minimizing physical restraints" as the use of the least restrictive method when physical restraints are required for inpatients. It is important to note that our focus was on older adult patients, excluding pediatric and psychiatric patients, given their distinct characteristics.

## Materials and methods

This research was conducted in three phases: (1) literature review, (2) interviews, and (3) the Delphi consensus process based on a prior indicator development process [16–19]. The potential indicators obtained from the literature review and interviews were organized inductively to develop draft indicators based on discussions among the researchers. Additionally, separate indicators were developed for top and middle management because of their distinct roles [20]. Top management included nurse executives and assistant nurse executives who typically assume leadership roles and participate in management based on their strategic and professional knowledge. They are potentially central to any new health service development or transformation [21]. Middle management included head nurses who utilize clinical governance processes at the ward level to ensure that patient care acts as a link between staff and top management nurses [22]. Therefore, two types of indicators were developed: top management and middle management.

### Phase 1: Literature review

We reviewed the nursing management literature on reducing physical restraints in acute hospitals to identify potential indicators [23]. Four databases, namely PubMed, CINAHL with Full Text, Cochrane Library, and Japan Medical Abstracts Society, were searched using the terms "Restraint, Physical" AND "management OR organize OR control OR Nursing, Supervisory" AND "acute hospital" NOT "Psychiatric Nursing OR Mental Health Nursing" on January 5, 2020. A total of 190 studies were identified in the search, of which 15 were considered

relevant to the theme of this study. In addition, we systematically reviewed existing guidelines and scientific literature regarding reducing physical restraints for older adults in acute care setting by referring to Japan Medical Abstracts Society [24] and searching on the internet [25–29]. A total of 192 potential indicators were extracted from textual data by trawling through existing guidelines and scientific literature related to the role of nursing management in minimizing physical restraints. These processes were discussed among authors to reach a consensus on the level of abstraction as potential indicators and appropriateness of text data selection through discussion.

## Phase 2: Interview

We interviewed four nurse executives and six head nurses whose experiences with reducing physical restraints were publicized on the internet [30] and who consequently identified 120 potential indicators. The interview codes were adopted as nursing management codes to minimize physical restraints. All three authors participated in the code extraction process, reaching a consensus on the level of abstraction as potential indicators and appropriateness of text data selection through discussion.

## Phase 3: The Delphi consensus process

We created a preliminary draft of indicators by categorizing and abstraction based on the 312 potential indicators from a point of view of differences and similarities which selected from Phases 1 and 2. Categorizing and abstraction by inductive content analysis were adopted because we lacked enough former knowledge about the phenomenon [31]. Top management had 35 draft indicators, while middle management had 33 to express the roles required for each level of management. A researchers' meeting was held to ensure that the indicators were appropriate for each level of management without compromising semantic content. Seven researchers, including the authors, attended this meeting, all of whom were experts in gerontological nursing and nursing management. The expert panelists were asked to assess the validity of each proposed indicator using a structured questionnaire, on a scale of 1–9, wherein 1 was *definitely invalid* and 9 was *definitely valid*. They could suggest revisions or additional indicators based on their experience and knowledge. The questionnaire was written in Japanese and reviewed by a certified nurse specialist in gerontological nursing (GCNS) and a geriatric nursing researcher, both of whom were not part of the expert panel or the researchers' meeting, before the Delphi consensus process. After assessment by the panelists, the researchers discussed each indicator for modification and adoption based on the panelists' rankings. The suggested indicators were reviewed for relevance and included in the next round of questionnaire based on researchers' consensus. The panelists received feedback on the previous round's results before assessing the modified draft indicators again. Each round lasted one month.

## Participants

The selection criteria for the panelists used in this Delphi study included having a specified number of relevant academic publications and professional experience/activity in the field of interest [19]. Therefore, we used convenience sampling to select panelists who could evaluate the draft indicators for minimizing physical restraints in older adults based on their rich knowledge or experience. Two categories of panelists were recruited; (1) certified nurse specialists recruited from the listed website of the Japan Academy of Gerontological Nursing and (2) nursing managers recruited from those who had been interviewed in a previous study [30] or reported nursing management practices about minimizing physical restraints on the internet. For each indicator, we selected 50 panelists who had extensive knowledge of or experience

in the theme of this study after reading about their respective activities posted on the internet. Eligible participants were mailed the study description and the questionnaire to their facility and enrolled only after they provided consent for participation. Those belonging to wards where older adults are not admitted, for example, pediatric and psychiatric wards, were excluded. The Delphi consensus process uses a minimum of 5–20 panelists and continues polling until the responses show stability; generally, three rounds are sufficient [19]. Additionally, the Delphi consensus process in the country studied had an average dropout rate of 42% over three rounds [32]. Based on previous studies, 50 panelists were recruited—25 GCNSs and 25 nursing managers for each indicator—to secure 5–20 panelists after three rounds in this study. Twenty-five nursing managers at each management level, including top and middle management, were recruited for convenience from those who reported nursing management practices about minimizing physical restraints and referred members of the same organizations using common selection methods in the Delphi consensus process [19]. Thus, panelists for top management indicators included 25 nurse executives or assistant nurse executives and 25 GCNSs, whereas those for middle management indicators included 25 head nurses and 25 GCNSs. This study was conducted between June and October 2021 in Japan.

## Measurement of study variables

**Draft management indicators to minimize physical restraints.** This study identified 35 draft indicators for top management and 33 for middle management. In developing these indicators, we considered the constructing metrics for this research theme based on the Scope and Standard of Nursing Administration Practice [33] and the Nursing Management Process [34]. Upon review, we established six metrics for basic nursing management activities in this research theme: (1) planning, (2) motivating, (3) training, (4) commanding, (5) organizing, and (6) controlling. The draft indicators for top management included six planning indicators, three motivating indicators, eight training indicators, four commanding indicators, ten organizing indicators, and four controlling indicators. The draft indicators for middle management included five planning indicators, six motivating indicators, six training indicators, eight commanding indicators, four organizing indicators, and four controlling indicators.

**Characteristics.** Participants' age, education level, position, whether GCNS or not, years of nursing experience, years of experience in an acute hospital, and years of nursing management experience (for nursing managers) were obtained.

**Analysis.** Descriptive statistics was used in each round of the Delphi consensus process. A median value of seven or higher, on a 9-point scale, served as the adaptation criterion for proposed indicators in subsequent rounds. Modifications and adoptions were made with reference to variance and free descriptions. Indicators were modified or rejected if the evaluation was divided. This occurred when at least three panelists rated them as having high validity (7, 8, 9) and low validity (1, 2, 3) in the same round, resulting in a lack of consensus. The consensus level, ranging from 1 to 9, was based on the RAND/UCLA Appropriateness Method, which assesses appropriateness for medical or surgical intervention [35]. This method was developed to synthesize the scientific literature and expert opinion on health care topics and used by the panelists to rate the benefit-to-harm ratio of the medical or surgical procedure on a scale of 1 to 9. This consensus level was used in the Delphi consensus process [16–18], which was appropriate for this study due to its good test-retest and interpersonal reliability, as well as good validity through comparison with guidelines [36]. Data were analyzed using SPSS 28.0 (IBM Corp., Armonk, NY, USA).

**Ethics.** This study was approved by the Institutional Review Board of the Medical Department of Yokohama City University on March 6, 2021 (approval no. 2021–002). Participants

filled out a self-administered paper questionnaire, which ensured quasi-anonymity through a correspondence table. All participants provided written informed consent before participating in each round of the study. We have not used any AI-assisted technologies in writing this article.

## Results

### Study population

In the first round of each process, 23 of the 50 panelists responded. In the second round, nine panelists from top management, and 10 from middle management dropped out. Finally, in the third round, 12 panelists remained in the top management process and 13 in the middle management process (Fig 1). The demographic characteristics of the participants are shown in Table 1. Most participants were in their 40s and had graduated from college or higher. Five staff nurses without management position participated in each management process; they were all GCNSs. Of the eight GCNSs, two were head nurses and one was an assistant nurse executive. The average number of years of nursing experience was 24.2 ± 8.8 in the top management process and 22.3 ± 4.6 in the middle management process, and the average number of years of experience in an acute hospital was 21.5 ± 9.9 in the top management process and 19.4 ± 7.6 in the middle management process. In a manager position, the mean number of years of nursing management experience was 14.3 ± 5.7 in the top management process and 10.8 ± 7.6 in the middle management process.

### Delphi consensus process

The Delphi consensus process for indicator development is depicted in Fig 2. The first round of the top management process inputted 35 indicators. There was no consensus on one indicator. We modified the 20 indicators that received comments, and two proposed indicators were added. Consequently, in the first round, 36 indicators were outputted and inputted to the next round. The second round found no consensus on one indicator. We modified the eight

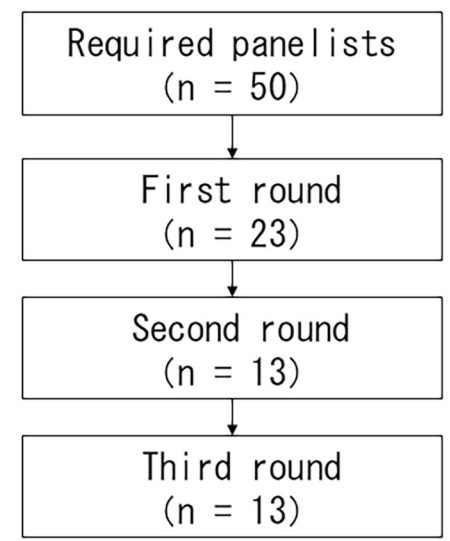

**Fig 1. Participant enrollment flowchart.**

**Table 1. Participant characteristics.**

| Variables | Top management process (n = 12) | Middle management process (n = 13) |
|---|---|---|
| **Age, years, n (%)** | | |
| **30s** | 3 (16.7) | 3 (23.1) |
| **40s** | 4 (22.2) | 6 (46.2) |
| **50s** | 5 (27.8) | 4 (30.8) |
| **Education level, n (%)** | | |
| **Technical school or Junior college** | 2 (25.0) | 5 (38.5) |
| **College or above** | 9 (75.0) | 8 (61.5) |
| **Position, n (%)** | | |
| **Staff nurse (all GCNSs)** | 5 (41.7) | 5 (38.5) |
| **Middle management position** | 2 (11.1) | 7 (53.8) |
| **Top management position** | 5 (41.7) | 1 (7.7) |
| **Other characteristics** | | |
| **GCNS, n (%)** | 8 (44.4) | 8 (61.5) |
| **Years of nursing experience, mean (SD)** | 24.2 (8.8) | 22.3 (4.6) |
| **Years of experience in acute hospitals, mean (SD)** | 21.5 (9.9) | 19.4 (7.6) |
| **Years of nursing management experience, mean (SD)** | 14.3 (5.7) (n = 7) | 10.8 (7.6) (n = 8) |

Abbreviations: GCNS, certified nurse specialist in gerontological nursing; SD, standard deviation

indicators that received comments. Thus, in the second round, 35 indicators were outputted and inputted to the next round. In the third round, all indicators met the criteria for a median score of seven or higher. We revised the 11 indicators slightly and received minor comments for more appropriate wording. The final dataset contained 35 indicators.

The first round of the middle management process inputted 33 indicators. There was no consensus on six indicators. We modified the 18 indicators that received comments, and one proposed indicator was added. Thus, in the first round, 28 indicators were outputted and inputted to the next round. The second round inputted 28 indicators. We modified the seven indicators that received comments. Consequently, in the second round, 28 indicators were outputted and inputted to the next round. In the third round, all indicators met the criteria for

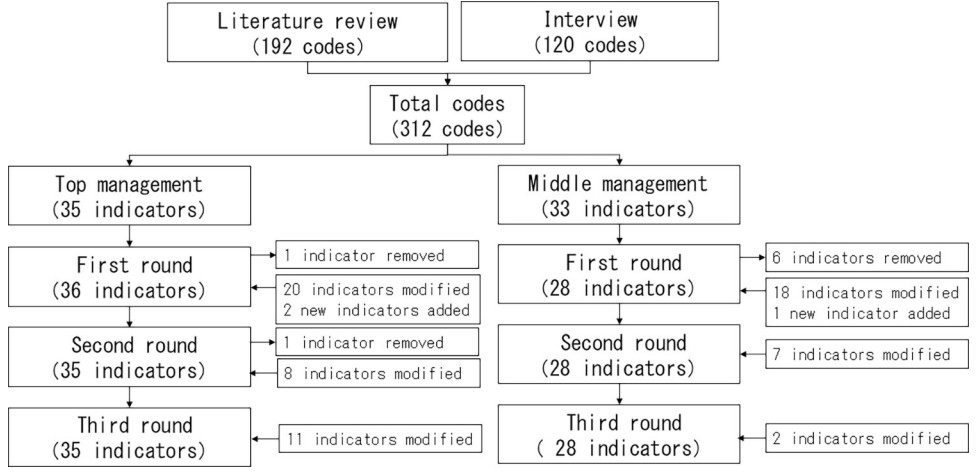

**Fig 2. Delphi consensus process for indicator development.**

a median score of seven or higher. We revised two indicators slightly and received minor comments for more appropriate wording. The final dataset contained 28 indicators.

Top management indicator no. 33 and middle management indicator no. 28 overlapped as nursing management activities were required for both. The final set of each indicator was however not dropped from the original six metrics: planning, motivation, training, commanding, organizing, and controlling. The management indicators of nursing for minimizing physical restraints (MaIN-PR) for top management are shown in Table 2, and those for middle management are shown in Table 3.

## Discussion

To our knowledge, this study provides the first set of nursing management indicators to minimize physical restraints, including a perspective on geriatric nursing in acute care settings. There is growing interest in nursing care indicators to assess the quality of nursing care in hospitals, including acute settings [37]. Previously, nursing care indicators have been drafted by the National Quality Forum to assess and improve the quality of nursing care in acute care hospitals [38]. Nursing management indicators are crucial for nursing care. The use of MaIN-PR to assess and improve nursing management activities would support a change in staff nurses' knowledge, perception, and attitude toward physical restraints. Additionally, the indicators provide guidelines for identifying nursing management activity. With demographic changes, older adult patients are set to become an increasingly important clientele in acute settings. MaIN-PR can serve as a mechanism to appraise nursing manager activities and an objective guideline to prepare high-quality nursing care for the aging population worldwide. Its strength lies in its ability to act as a guideline for nursing managers in an acute setting and identify specific nursing managers' activities to minimize physical restraints.

### Validity of MaIN-PR items

We developed 35 indicators for top management and 28 indicators for middle management. Suggested guidelines for using the Delphi method [19], particularly when relatively little prior research has been conducted in the field or when the survey needs to be exhaustive in scope, indicate that 50–85 draft indicators may be needed. Prior studies have also developed similar extensive indicators (e.g., 79 indicators [16], 90 indicators [17], and 35 indicators [18]). Therefore, we believe we have created a reasonable number of indicators and provide a helpful nursing management perspective for minimizing physical restraints in practice.

In addition, some indicators were removed during the Delphi consensus process; however, six metrics were maintained in the top and middle management indicator structures: (1) planning, (2) motivation, (3) training, (4) commanding, (5) organizing, and (6) controlling.

Planning was composed of indicators such as the analysis of the situation, setting goals and policies, and the creation of standards. These indicators focused on nursing managers as powerful influential operators, which is a role that has been identified in previous studies [39].

Motivation comprised indicators, such as the motivation of individuals and support for goal-directed actions. These indicators are important items in the MaIN-PR for each staff nurse to actively pursue direct care to minimize physical restraints. Prior studies have also indicated that first-line nursing managers must motivate nursing employees [40]. Additionally, in this study, several indicators were placed in middle management, which directly involves staff nurses.

The training consisted of indicators related to staff education. An educational theme on geriatric nursing was adopted as the basis for minimizing physical restraints. The main

**Table 2. Comparison of results of management indicators of nursing for minimizing physical restraints for panelists from top management before the third round.**

| No. | Draft indicator | Final set indicator | 1–3 (n) | 7–9 (n) | Median |
|---|---|---|---|---|---|
| **Planning** | | | | | |
| 1 | Incorporating findings on minimizing physical restraints outside of hospitals and on the environment surrounding acute care hospitals | Gathering information on the environment surrounding acute care hospitals and findings on minimizing physical restraints outside of hospitals | 0 | 10 | 8 |
| 2 | Visualization of issues related to physical restraints occurring in the nursing department | Visualization of issues related to physical restraints occurring in the field | 0 | 12 | 8.5 |
| 3 | Statement of policy to minimize physical restraints | Statement of policy to minimize physical restraints at the hospital or nursing department | 0 | 12 | 9 |
| 4 | Instilling the policy of minimizing physical restraints in the nursing department | Spreading the policy of minimizing physical restraints throughout the nursing department | 0 | 12 | 9 |
| 5 | Clarification on the definition of physical restraint | Clarification on the definition of physical restraint | 0 | 12 | 9 |
| 6 | Providing standards in the hospital to determine the need for physical restraints | Providing reference standards in the hospital to determine the need for physical restraints | 0 | 12 | 9 |
| 7 | Creating manuals or guidelines that address the factors that lead to the implementation of physical restraints | Utilization of existing manuals or guidelines or creating ones that address the factors that lead to the implementation of physical restraints at each hospital | 0 | 12 | 9 |
| **Motivating** | | | | | |
| 8 | Sharing success experiences about minimizing physical restraints within the nursing department | Sharing success experiences about minimizing physical restraints within the nursing department | 0 | 12 | 9 |
| 9 | Sharing the reactions of patients and families to the implementation of nursing care to minimize physical restraints within the nursing department | Sharing the reactions of patients and families to the implementation of nursing care to minimize physical restraints within the nursing department | 0 | 12 | 8.5 |
| 10 | Being an organizational culture to reduce physical restraints | Attempting to create an organizational culture with daily initiatives and discussions toward minimizing physical restraints | 0 | 12 | 9 |
| **Training** | | | | | |
| 11 | Planning for required education to minimize physical restraints based on the actual situation and existing evidence of physical restraints | Planning for required education to minimize physical restraints concerning the actual situation and existing evidence of physical restraints | 0 | 12 | 8 |
| 12 | Assigning staff in charge of promoting the minimization of physical restraints at each department | Planning to develop leaders who will respect the dignity of patients and improve their ethical competence | 0 | 11 | 9 |
| 13 | Educating the staff to spread the correct knowledge and skills necessary for minimizing physical restraints | Providing opportunities to spread the correct knowledge and skills necessary to not implement physical restraint whenever possible | 1 | 11 | 9 |
| 14 | Educating the staff to acquire skills for minimization of physical restraints | Maintenance of an educational system for basic older adult patients' care and delirium care | 0 | 12 | 9 |
| 15 | Researching efforts to minimize physical restraints | Recommending research-based efforts to minimize physical restraints | 0 | 10 | 8 |
| 16 | Adopting educational tools to ensure that all staff members receive education concerning minimizing physical restraints | Adopting educational tools to ensure that all staff members receive necessary education for minimizing physical restraints | 1 | 11 | 8 |
| 17 | Encouraging staff voluntary growth and helping it reflect on daily ethics and usual nursing practice regarding physical restraints from an ethical perspective | Recommend creating opportunities to reflect on daily ethics and usual nursing practice with a focus on patient intention and quality of life | 0 | 12 | 9 |
| **Commanding** | | | | | |
| 18 | Survey and analysis of staff's awareness of physical restraints | Confirming the staff's feelings and awareness of respect for dignity in minimizing physical restraints, as perceived through daily interactions | 0 | 11 | 8 |
| 19 | Sharing with each department the discussions among managers and committees regarding physical restraints to review the ethical aspect | Sharing with each department the discussions and improvement measures among managers and committees regarding physical restraints | 0 | 12 | 8.5 |
| 20 | Intentionally involving individuals in a leadership position in the department: the head nurse and chief nurse | Proactively interacting with individuals in a leadership position in the department: the head nurse and chief nurse, and involving them in efforts to minimize physical restraints | 0 | 12 | 9 |
| 21 | Considering the impact of no use of physical restraints on hospital management and sharing with other departments and nursing departments | Supporting each department's efforts to minimize physical restraints from a managerial perspective, and discussing the direction with the organization's executive managers | 0 | 10 | 8 |
| **Organizing** | | | | | |

*(Continued)*

**Table 2.** (Continued)

| No. | Draft indicator | Final set indicator | 1–3 (n) | 7–9 (n) | Median |
|---|---|---|---|---|---|
| 22 | Placing cross-organizational teams to promote minimizing physical restraints | Using committees and teams to promote minimizing physical restraints | 0 | 12 | 9 |
| 23 | Making efforts to minimize physical restraints by multidisciplinary teams or multiple staff | Supporting efforts to minimize physical restraints by multidisciplinary teams | 0 | 11 | 8 |
| 24 | Considering care methods to cooperate with staff in each department and cross-sectional teams | Considering care methods to cooperate with staff in each department and cross-sectional teams | 0 | 11 | 8.5 |
| 25 | Creating a system to consult with specialists in the hospital regarding physical restraints | Creating of a system to consult with specialists in the hospital regarding physical restraints | 0 | 11 | 9 |
| 26 | Did not exist | Creating a mechanism to consult with teams or specialists when ethical issues arise | 0 | 11 | 8 |
| 27 | Establishing a system to share ethical issues related to physical restraints with managers, each committee, and each medical department | Sharing ethical issues related to physical restraints with other departments, including each medical department, and establishing a system of collaboration that allows multidisciplinary dialogue | 0 | 11 | 8 |
| 28 | Devising a working system to look after patients | Establishing a flexible cross-departmental support system to ensure that staff are available to care for patients | 0 | 11 | 8 |
| 29 | Showing attitude to guarantee responsibility against accidents associated with minimizing physical restraints | Guaranteeing organization responsibility against accidents associated with minimizing physical restraints | 0 | 11 | 9 |
| 30 | Did not exist | Creating mechanisms and opportunities for patients and families to understand the minimizing physical restraints as an organization | 0 | 12 | 8 |
| 31 | Creating an environment, maintenance, and management of supplies to prevent accidents | Creating an environment, maintenance, and management of supplies to prevent accidents | 0 | 12 | 8.5 |
| | Placing a committee to review issues related to physical restraints | [delete] (integrating with no. 23) | | | |
| | Creating opportunities to reflect on nursing practices from the patient's perspective to foster an ethical view of physical restraints | [delete] (integrating to no. 14) | | | |
| **Controlling** | | | | | |
| 32 | Ongoing evaluation of the effort progress minimizing physical restraints at your hospital | Ongoing evaluation of the effort progress minimizing physical restraints at your hospital | 0 | 12 | 9 |
| 33 | Survey and analysis of the number and percentage of physical restraints in the hospital | Periodic survey and analysis of indicators, such as the number and percentage of physical restraints used and fall rates | 0 | 12 | 8.5 |
| 34 | Analysis of physical restraint rates by comparing with external evaluation criteria | Analysis of physical restraint rates concerning external evaluation criteria | 0 | 11 | 7 |
| 35 | Reflecting for practice from the results of the survey analysis for physical restraints | Identifying issues in work processes based on the results of the survey analysis for physical restraints and discussing improvement measures together with each department | 0 | 10 | 8 |

indicator for top management was planning for training, whereas for middle management, staff nurses were encouraged to participate in training.

Commanding comprised indicators such as handling conflicts, problem-solving, and communication. The role of executive nurses is effective communication [39], and first-line nurse managers' managerial competencies are indicated by communicating organizational goals, managing conflicts, and solving problems [40]. Staff nurses face an ethical dilemma regarding physical restraints between the characteristics of nursing care of older adult patients, ensuring safety, and carrying out treatment [10]. Thus, communication with staff nurses and managing conflict is an important nursing management activity to solve problems related to physical restraints.

Organization encompassed indicators such as group work, team building, and the clarification of responsibilities. A nurse executive director's role is to provide nurses with the right tools and resources to perform their jobs [39]. This indicator showed the contents needed to create resources to facilitate staff nurses' positive approach to minimizing physical restraints. MaIN-PR for top management holds more indicators in "organizing" than middle management because this category ensures that the role of top management is fully reflected.

**Table 3. Comparison of results of management indicators of nursing for minimizing physical restraints for panelists from middle management before the third round.**

| No. | Draft indicator | Final set indicator | 1–3 (n) | 7–9 (n) | Median |
|---|---|---|---|---|---|
| **Planning** | | | | | |
| 1 | Visualization of issues related to physical restraints in the organization | Visualization of issues related to physical restraints in the department | 0 | 12 | 9 |
| 2 | Developing the departmental targets to understand the nursing department's policy to minimize physical restraints | Developing and spreading departmental targets to understand the nursing department's policy to minimize physical restraints | 0 | 13 | 9 |
| 3 | Documenting standards for staff to determine the need for physical restraints | Providing reference standards to determine the need for physical restraints that staff can consult | 0 | 12 | 8 |
| 4 | Grasping the status of efforts to minimize physical restraints | Grasping the status of efforts to minimize physical restraints based on committee and departmental discussions | 0 | 12 | 9 |
| 5 | Encouraging staff to take advantage of problem-solving opportunities with multidisciplinary to minimize physical restraints | Striving to participate in discussions with physicians and other professionals to minimize physical restraints, and understand the results of these discussions | 0 | 12 | 8 |
| | Communicating to staff the policy of minimizing physical restraints | [delete] (integrating with no. 2) | | | |
| **Motivating** | | | | | |
| 6 | Admitting staff's positive attitude toward minimizing physical restraints | Positive evaluation of staff's positive attitude toward minimizing physical restraints | 0 | 13 | 9 |
| 7 | Did not exist | Feedback recognizing the good points and autonomy of efforts to minimize physical restraints for staff on a daily basis | 0 | 12 | 9 |
| 8 | Feedback on what is discussed about physical restraints at committee meetings and in administrative departments for staff | Feedback on what is discussed about physical restraints at committee meetings and in administrative departments for staff | 0 | 12 | 8 |
| 9 | Improving staff self-efficacy by sharing patient/family responses to staff who have implemented nursing care to minimize physical restraints | Improving staff self-efficacy by returning patient/family responses to staff who have implemented nursing care to minimize physical restraints | 0 | 12 | 9 |
| 10 | Sharing success experiences for minimizing physical restraints | Creating opportunities for staff to share success experiences for minimizing physical restraints | 0 | 13 | 9 |
| **Training** | | | | | |
| 11 | Providing opportunities or encouraging education to spread the correct knowledge and skills about physical restraints | Providing opportunities or encouraging participation in education to spread the correct knowledge and skills necessary to not implement physical restraint whenever possible | 0 | 13 | 9 |
| 12 | Providing opportunities or encouraging participation in education as a base for older adult patients' care and delirium care | Providing opportunities or encouraging participation in education as a base for older adult patients' care and delirium care | 0 | 13 | 9 |
| 13 | Enabling reflection on nursing care from the patient's point of view to foster an ethical view of physical restraints | Setting up opportunities to reflect on nursing care from the patient's viewpoint to foster an ethical view of physical restraints | 0 | 13 | 9 |
| 14 | Enhancing to reflect on usual nursing practice from the perspective of ethics | Providing opportunities to reflect on daily ethics and usual nursing practice with a focus on patient intention and quality of life | 0 | 13 | 9 |
| 15 | Research practice to minimize physical restraints | Active action for research-based practice and practice reporting about successful cases to minimize physical restraints | 0 | 11 | 8 |
| | Recommending the use of educational tools to ensure that all staff receive education on physical restraints | [delete] (integrating nos. 12 and 13) | | | |
| **Commanding** | | | | | |
| 16 | Setting up a discussion forum for multiple staff members to discuss minimizing physical restraints | Setting up a discussion forum for multiple staff members to discuss minimizing physical restraints | 0 | 13 | 9 |
| 17 | Providing opportunities for dialogue with patients and families regarding physical restraints to gain their understanding and cooperation in minimizing physical restraints | Providing opportunities for dialogue with patients and families regarding physical restraints to gain their understanding and cooperation in minimizing physical restraints | 1 | 12 | 8 |
| 18 | Considering care methods for factors that contribute to physical restraints with cross-functional teams and specialists | Considering care methods for factors that contribute to physical restraints with cross-functional teams and specialists | 0 | 12 | 9 |
| 19 | Discussing with other professionals and specialists about ethical dilemmas regarding physical restraints | Discussing with other professionals and specialists about ethical dilemmas regarding physical restraints | 1 | 12 | 9 |
| 20 | Sharing ethical issues related to physical restraints with staff | Sharing ethical issues related to physical restraints with staff | 0 | 13 | 9 |
| 21 | Collaborating with staff to determine care alternatives to physical restraints in the field | Work with staff to determine care alternatives to physical restraints when staff members are struggling | 0 | 13 | 9 |

*(Continued)*

**Table 3.** (Continued)

| No. | Draft indicator | Final set indicator | 1–3 (n) | 7–9 (n) | Median |
|---|---|---|---|---|---|
| 22 | Use of manuals and guidelines to address factors when physical restraints adopted | Use of manuals and guidelines to address factors when physical restraints adopted by the facility | 1 | 11 | 8 |
| 23 | Negotiation with administrative departments on providing necessary environmental arrangements to minimize physical restraints | Negotiation with administrative departments on providing necessary environmental arrangements to minimize physical restraints | 1 | 11 | 9 |
|  | Involving the chief and specialists to facilitate staff discussions about care and minimizing physical restraints | [delete] (integrating with no. 21) |  |  |  |
| **Organizing** | | | | | |
| 24 | Supporting the activities of staff that promote minimizing physical restraints | Supporting the activities of staff that promote minimizing physical restraints | 0 | 13 | 8 |
| 25 | Trying to discuss the point of removal of physical restraints | Discussing led by department leaders about alternatives to physical restraints and the practice of dealing with patients | 0 | 11 | 9 |
| 26 | Responding to reduce fear about accidents associated with minimizing physical restraint with an attitude of accepting responsibility | Responding to prevent individual fear of accidents associated with minimizing physical restraint through team discussions | 0 | 12 | 9 |
| 27 | Prepare the accident prevention items and restraint substitutes | Maintenance of department-owned accident prevention items and restraint substitutes for available use when needed | 0 | 13 | 8 |
| **Controlling** | | | | | |
| 28 | Survey and analysis of the number and percentage of physical restraints | Periodic survey and analysis of indicators, such as the number and percentage of physical restraints used and fall rates | 1 | 12 | 8 |
|  | Survey and analysis of staff's awareness of physical restraints | [delete] (lack of consensus) |  |  |  |
|  | Analysis of physical restraint rates with comparing to external evaluation criteria | [delete] (lack of consensus) |  |  |  |
|  | Reflecting for practice from the results of the survey analysis for physical restraints | [delete] (lack of consensus) |  |  |  |

Controlling comprised indicators to evaluate the plan and lead to the next "planning" activity. The content included understanding the status of the hospital as a whole and comparing it with those of other hospitals for top management, and understanding the status of their departments for middle management. They were separated according to the role of each management level.

Indicators are available for improving patient outcome [41]. MaIN-PR could indirectly turn around negative outcomes from physical restraints [6–9] and contribute to positive outcomes, such as quality of life improvement, decreased mortality, and shorter hospital stays. Prior studies investigated organizational interventions aimed at implementing a minimizing physical restraint policy; however, they found no clear evidence that they are effective at reducing physical restraint use for hospitalized older adult patients [42]. MaIN-PR could help to plan further interventions on effective organizational strategies to minimize physical restraints for older adult patients hospitalized in acute care settings.

## Validity of the Delphi consensus process in MaIN-PR development

By developing MaIN-PR using the Delphi consensus process, we established indicators with evidence that obtained a unified expert opinion on nursing management to minimize physical restraints in older adult patients in acute care settings, which has not yet been clarified.

This study had 5–20 experts (top management: 12, middle management: 13) who participated in three rounds to reach a consensus on the indicators [19]. The Delphi consensus process, based on the assumption of safety in numbers and the validity of that judgment, was reinforced by a critical discussion of the assumptions [43]. Thus, MaIN-PR ensured validity by considering consensus and disagreement in expert panel opinions through three rounds of the

Delphi consensus process. The median number of years of experience in the specialty area was 15 years in a previous Delphi consensus process [44]. We recruited appropriate participants for expert panels for top management because years of experience in acute hospitals (21.5 ± 9.9 years) and nursing management experience (14.3 ± 5.7 years) in top management were more than 15 years. However, years of nursing management experience in middle management were lower than 15 years (10.8 ± 7.6 years). Expert panels in the Delphi consensus process tend to be used for purposeful or convenience sampling [45]. Head nurses were invited to participate by introducing an executive nurse. Experience with working on strategies to minimize physical restraint should be prioritized over experience in nursing management.

## Limitations

The first limitation is related to decreased participation. The number of participants decreased from the first round to second and third rounds. While it is known that greater the number of rounds, greater is the degree of dropout in the Delphi consensus process [19], retention of a greater number of participants would have made the results more robust. Moreover, response bias may have occurred because fewer people participated in the second and third rounds, as well as based on recalling past events. The second limitation concerns the sampling method. We used convenience sampling that may have affected the generalizability of our sample. The final limitation relates to unreported practical testing. Further testing of MaIN-PR for reliability, feasibility, and usability in practice is required including the number of indicators.

## Conclusion

This study aimed to develop MaIN-PR by using the Delphi consensus process to establish evidence-based basic standards. The nursing managers and certified nurse specialists in gerontological nursing were asked to assess the validity of each indicator in three rounds. We developed 35 indicators for top management and 28 indicators for middle management, which were classified into six metrics: planning, motivating, training, commanding, organizing, and controlling. To our knowledge, MaIN-PR is the first set of nursing management indicators that aims to minimize physical restraints. Further testing of MaIN-PR for reliability, feasibility, and usability in practice is required.

## Supporting information

**S1 Table. Guidelines for Conducting and Reporting Delphi Studies (CREDES).**
(DOCX)

**S1 Appendix. Expert panel assessment of indicators in the first to third rounds.**
(DOCX)

**S2 Appendix. Questionnaire used for top managers in the first round.**
(PDF)

**S3 Appendix. Questionnaire used for middle managers in the first round.**
(PDF)

## Acknowledgments

We thank Professor Azusa Arimoto for insightful comments and the participants who completed the questionnaires.

## Author Contributions

**Conceptualization:** Maya Minamizaki.

**Data curation:** Maya Minamizaki.

**Formal analysis:** Maya Minamizaki, Mana Doi, Yuka Kanoya.

**Funding acquisition:** Maya Minamizaki.

**Investigation:** Maya Minamizaki.

**Methodology:** Maya Minamizaki.

**Project administration:** Yuka Kanoya.

**Supervision:** Mana Doi, Yuka Kanoya.

**Writing – original draft:** Maya Minamizaki.

**Writing – review & editing:** Mana Doi, Yuka Kanoya.

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
