## [Decision Letter · Decision Letter 0]

22 Apr 2024

PONE-D-23-31249Development of management indicators of nursing for minimizing physical restraints focused on older adult patients hospitalized in acute care settings: a Delphi consensus studyPLOS ONE

Dear Dr. Minamizaki,

Thank you for submitting your manuscript to PLOS ONE. After careful consideration, we feel that it has merit but does not fully meet PLOS ONE’s publication criteria as it currently stands. Therefore, we invite you to submit a revised version of the manuscript that addresses the points raised during the review process.

We look forward to receiving your revised manuscript.

Kind regards,

Alejandro Botero Carvajal, MD

Academic Editor

PLOS ONE

Journal Requirements:

Additional Editor Comments:

The article is relevant, but requires specifying key aspects at a methodological level; it is advisable to follow each of the annotations made to improve the quality of the manuscript.

Reviewers' comments:

Reviewer's Responses to Questions

**Comments to the Author**

1. Is the manuscript technically sound, and do the data support the conclusions?

Reviewer #1: Partly

Reviewer #2: Yes

2. Has the statistical analysis been performed appropriately and rigorously? 

Reviewer #1: N/A

Reviewer #2: No

3. Have the authors made all data underlying the findings in their manuscript fully available?

Reviewer #1: Yes

Reviewer #2: No

4. Is the manuscript presented in an intelligible fashion and written in standard English?

Reviewer #1: Yes

Reviewer #2: Yes

5. Review Comments to the Author

Reviewer #1: Thank you very much for the opportunity to review the interesting manuscript on a relevant topic. I have read the manuscript with interest, following you find my comments

Abstract

The abstracts comprise important information, but the structure can be improved. Please add details on the design and methods (please see my comments below on this topic).

Please add information on how nursing management indicators and nursing management activities are linked and how they are expected to have an impact on clinical practice.

“The basic standard for nursing management activities to minimize physical restraints in acute care settings requires evidence of quality concerns”. Please revise this sentence, the meaning is not entirely clear to me.

Please add information on the number of participants responding in the different rounds of the Delphi.

Manuscript

Background

I recommend using the consensus definition on physical restraints (Bleijlevens et al. Physical Restraints: Consensus of a Research Definition Using a Modified Delphi Technique. J Am Geriatr Soc. 2016;64:2307-10. doi: 10.1111/jgs.14435).

The background section describes important aspects of the topic, but partly lacks conceptual clarity.

Please define “nursing management indicators”.

Please clarify what is meant with “nursing management activities to minimize physical restraints”.

The contextual information about the use of restraints should be sharpened. For example, “In particular, older adult patients receiving acute care are more likely to need physical restraints owing to the high risk of falls and difficulty in continuing treatment because of confusion based on cognitive and physical decline” might indicate that there is a “need” for restraint use in some people in acute care based on specific characteristics. However, restraints are not effective to reduce falls or prevent unplanned extubation. The negative impact is mentioned in the background, but the sentence mentioned above and the following sentence should be revised to increase clarity.

“However, nurses face an ethical dilemma about physical restraints between the characteristics of nursing care for older adult patients, ensuring their safety, and providing treatment [13] making it difficult for them to make decisions to avoid physical restraints.”

It is not clear to me why the strategies to reduce restraints were mentioned, as there is no convincing evidence on their effects. If I understand the authors correctly, this is the reason why they used a different approach. This should be clarified.

Please add information on how the indicators are expected to lead to changes in clinical practice regarding the use of physical restraints. Please describe the expected mechanism of change.

“Therefore, we identified indicators to determine the basic standard of nursing management for minimizing physical restraints.” Please revise this sentence to increase clarity.

Please check the background section for redundancies.

Design and methods

Delphi is a method, but no design.

Please provide a general overview of the different phases including basic information on design and methods.

Phase 1 and 2 are not described in sufficient detail.

Phase 3

Please describe in more detail the “inductive process” used to develop the preliminary draft of indicators based on the Phases 1 and 2.

Please describe the expert panel in more detail (inclusion criteria, recruitment, characteristics and qualifications of participants). Was the Delphi conducted with the expert panel? It might be helpful to restructure the description of Phase 3.

Please add some explanation about “top management” and “middle management”. Please also clarify whether the GCNSs refer to the middle management and nursing managers refer to the top management. If so, please revise the manuscript and use the same terms to increase clarity.

Results

Study population

Please check Figure 1, it appears that 100 people were initially on the panel list, but earlier it was described that 25 GCNSs and 25 nursing managers were invited.

Description of the Delphi process: please check the number of indicators presented in the text and in Figure 2 for the different rounds. For example, middle management: in the text it is described that 28 indicators were included in the first round, in Figure 2 the number is 33.

There seems to be some overlap in the indicators for middle and top management as well as within the two sets of indicators. I recommend adding some information on this topic in the results section.

Please also comment on whether the feasibility or usability of the indicators was addressed in the Delphi.

Discussion

The number of indicators is very high and the use of physical restraint is only one topic beyond others important. Please discuss the feasibility of such an extensive list of indicators and the practical implications.

Reviewer #2: It is significant, valuable study, as it addresses an important and ethical issues in missed area (old age care).

- When the population framework is known it is mandatory to use the probability sampling technique, and you need to state how you select the sample size

another important an ethical issue the researcher have to consider is using AI in writing scientific paper

also special consideration to the references, old ref should not be used.

6. PLOS authors have the option to publish the peer review history of their article (what does this mean?). If published, this will include your full peer review and any attached files.

Reviewer #1: No

Reviewer #2: No

---

## [Author Response · Author response to Decision Letter 0]

4 Jun 2024

Dear Dr. Carvajal:

Thank you for giving us the opportunity to submit a revised draft of our manuscript and for your comments supporting the significance of this study in the geriatric nursing field. We also appreciate the time and effort that you and the reviewers dedicated to providing feedback on our manuscript. We agree with most of the suggestions made by you and the reviewers and have revised our manuscript accordingly. Any overlap between your comments and those by the reviewers has been noted in our response to the reviewers. To ensure the newly added text is easy to identify, we have highlighted the changes in yellow within the manuscript.

Editor’s Comments to the Authors:

Introduction

1. It is very concise and it is preferable to write a little about role of nurse’s advocacy as it important in reducing using of restrain particularly among vulnerable frail elderly.

Author Response: Thank you for pointing this out. As suggested by the editor, we have added the need for nursing advocacy in minimizing physical restraints in the revised manuscript. Please see page 3, lines 42–44.

Materials and methods

2. Use the term indicators instead of (codes) or vis versa: as it preferable to use the same term.

Author Response: Thank you for your suggestion. The term “code” has been replaced with “potential indicator” throughout the revised manuscript.

Analysis

3. Add the finding of RAND/UCLA Appropriateness Method 

Author Response: We appreciate your comment. We have added an explanation of the RAND/UCLA Appropriateness Method in our revised manuscript. Please see page 9, lines 166–170.

4. Researcher need to perform the reliability test: to use these indicators to minimize physical restraints again

Author Response: Thank you for your comments. This study did not include a test of reliability. This has now been included in the revised manuscript as a limitation of this study. Please see page 34, lines 322–324.

Table1

5. Revise the categories of participants in table 1: staff nurses present in this table, and in the rest of study there are two categories: Middle management position and top management position

Author Response: Thank you for your comments. Five staff nurses without management position participated in each management process; they were all certified nurse specialists in gerontological nursing (GCNSs). For ease of understanding, we have included this information in the text and the tables.

Limitations

6. Well written, but the researcher needs to add sampling technique used as one of limitations.

Author Response: We appreciate your comment. As suggested by the editor, we have included the lack of clear generalizability as one of the study limitations. Please see page 34, lines 321–322.

Conclusion

7. Rewrite it again: Summarizes the main points and suggests implications for future research.

Author Response: Thank you for your comments. As suggested by the editor, we have rewrite conclusion for clarity and conciseness. Please see page 34–35, lines 326–333.

Reviewers’ Comments to the Authors:

Reviewer 1

Abstract

The abstracts comprise important information, but the structure can be improved. 

1. Please add details on the design and methods (please see my comments below on this topic). 

2. Please add information on how nursing management indicators and nursing management activities are linked and how they are expected to have an impact on clinical practice

3. “The basic standard for nursing management activities to minimize physical restraints in acute care settings requires evidence of quality concerns”. Please revise this sentence, the meaning is not entirely clear to me.

4. Please add information on the number of participants responding in the different rounds of the Delphi

Author Response: Thank you for your comments. As suggested by the reviewer, we have added explanations of this study’s framework and information on the number of participants in each round of the Delphi. We have also revised the unclear sentence pointed out by the reviewer. Please see page 2, lines 14–16 and 26–27.

Background

5. I recommend using the consensus definition on physical restraints (Bleijlevens et al. Physical Restraints: Consensus of a Research Definition Using a Modified Delphi Technique. J Am Geriatr Soc. 2016;64:2307-10. doi: 10.1111/jgs.14435) 

The background section describes important aspects of the topic, but partly lacks conceptual clarity. Please define “nursing management indicators”. Please clarify what is meant with “nursing management activities to minimize physical restraints”.

Author Response: Thank you for pointing this out. As suggested by the reviewer, we have adopted the consensus definition on physical restraints (Bleijlevens et al., 2016) because it includes our understanding of physical restraints. In addition, we have clarified the definitions of “nursing management indicators” and “minimizing physical restraints.” Please page 3, lines 36–38 and page 4, lines 58–63.

6. The contextual information about the use of restraints should be sharpened. For example, “In particular, older adult patients receiving acute care are more likely to need physical restraints owing to the high risk of falls and difficulty in continuing treatment because of confusion based on cognitive and physical decline” might indicate that there is a “need” for restraint use in some people in acute care based on specific characteristics. However, restraints are not effective to reduce falls or prevent unplanned extubation. The negative impact is mentioned in the background, but the sentence mentioned above and the following sentence should be revised to increase clarity.

“However, nurses face an ethical dilemma about physical restraints between the characteristics of nursing care for older adult patients, ensuring their safety, and providing treatment [13] making it difficult for them to make decisions to avoid physical restraints.”

It is not clear to me why the strategies to reduce restraints were mentioned, as there is no convincing evidence on their effects. If I understand the authors correctly, this is the reason why they used a different approach. This should be clarified.

Author Response: We agree with the reviewer’s assessment. Your suggestion had provided our background with considerable academic insight and clarity. While making decisions and taking actions to remove physical restraints, nurses may be under the misconception that physical restraints are safe. Therefore, they need support in minimizing physical restraints. We have clarified as to why we need strategies to minimize physical restraints. We have also revised the relevant text in the introduction section of the revised manuscript. Please see page 3, lines 38–40 and 42–45.

7. Please add information on how the indicators are expected to lead to changes in clinical practice regarding the use of physical restraints. Please describe the expected mechanism of change. “Therefore, we identified indicators to determine the basic standard of nursing management for minimizing physical restraints.” Please revise this sentence to increase clarity. Please check the background section for redundancies.

Author Response: Thank you for your comments and suggestions. We have clarified the mechanism that leads nurses to use physical restraints. We believe that MaIN-PR would impact nurses’ knowledge, perception, and attitude toward physical restraints in clinical practice. Please see page 3, lines 46–49.

Design and methods

8. Delphi is a method, but no design. Please provide a general overview of the different phases including basic information on design and methods.

Phase 1 and 2 are not described in sufficient detail.

Phase 3

Please describe in more detail the “inductive process” used to develop the preliminary draft of indicators based on the Phases 1 and 2.

Author Response: Thank you for your comments. We received similar comments from the editor. In response, we have expanded our description of the methods used to obtain the codes from the literature and interviews and the analysis methods used in Phase 3. Please see page 4, lines 65–66; page 5, lines 77–89 and 91–92; and page 6, 98–103.

9. Please describe the expert panel in more detail (inclusion criteria, recruitment, characteristics and qualifications of participants). Was the Delphi conducted with the expert panel? It might be helpful to restructure the description of Phase 3.

Please add some explanation about “top management” and “middle management”. Please also clarify whether the GCNSs refer to the middle management and nursing managers refer to the top management. If so, please revise the manuscript and use the same terms to increase clarity.

Author Response: We appreciate your comments. We were also asked by the editor to explain our decision to use the convenience sampling technique and the inclusion and exclusion criteria. The panelists were nursing managers and GCNSs who had experience in or knowledge of minimizing physical restraints in older adults in acute care setting. Participants were selected using the convenience sampling method because the sensible panelist selection criteria used in Delphi research include having a specified number of relevant academic publications and professional experience/activity in the field of interest (Belton et al., 2019). We have clarified the selection and composition of the panelists evaluating the top and middle management indicators. Please see pages 7, lines 118–122 and 129–130.

Belton I, MacDonald A, Wright G, Hamlin I. Improving the practical application of the Delphi method in group-based judgment: A six-step prescription for a well-founded and defensible process. Technol Forecasting Soc Change. 2019;147: 72-82. doi: 10.1016/j.techfore.2019.07.002.

Result

10. Please check Figure 1, it appears that 100 people were initially on the panel list, but earlier it was described that 25 GCNSs and 25 nursing managers were invited.

Author Response: Thank you for your comments, and we apologize for our earlier explanation that was unclear and misleading. Panelists of top management indicators included 25 nurse executives or assistant nurse executives and 25 GCNSs. Panelists of middle management indicators included 25 head nurses and 25 GCNSs. We have accordingly modified the methods section. Please see page 7–8, lines 126–129 and 139–141.

11. Description of the Delphi process: please check the number of indicators presented in the text and in Figure 2 for the different rounds. For example, middle management: in the text it is described that 28 indicators were included in the first round, in Figure 2 the number is 33.

Author Response: Thank you for your comments, and we apologize for our earlier explanation that was unclear and misleading. In the first round of middle management indicator evaluation, panelists assessed 33 indicators, which were revised to 28 indicators after the ratings were tallied. We realize that the description of inputted and outputted indicators for each round was unclear in the original manuscript. We have therefore modified the results section and Figure 2 in the revised manuscript. Please see page 12, lines 201–220.

12. There seems to be some overlap in the indicators for middle and top management as well as within the two sets of indicators. I recommend adding some information on this topic in the results section.

Please also comment on whether the feasibility or usability of the indicators was addressed in the Delphi.

Author Response: Thank you for your comments, and we apologize for our earlier explanation that was unclear and misleading. We have added a statement in the revised manuscript that explains that top management indicator no. 33 and middle management indicator no. 28 were duplicates. This study did not include tests of feasibility and usability. This has been included in the limitations section of the revised manuscript. Please see page 13, lines 221–223.

Discussion

13. The number of indicators is very high and the use of physical restraint is only one topic beyond others important. Please discuss the feasibility of such an extensive list of indicators and the practical implications.

Author Response: The reviewer has raised some very important points; however, we do not believe the number of indicators is extremely high based on prior developed indicators. Suggested guidelines for the Delphi method (Belton et al., 2019) state that in cases where relatively little prior research has been conducted in the field or where the survey needs to be exhaustive in scope, 50–85 draft indicators may be needed. Doody et al. (2019) developed 12 nursing quality care process metrics and 79 indicators for intellectual disability services. Murphy et al. (2019) developed 20 nursing quality care process metrics and 90 indicators for use in care settings for older persons. Hamatani et al. (2020) developed 35 indicators about palliative care in patients with chronic heart failure. We, however, developed six metrics and 33 or 28 indicators with different targets to be answered. The number of indicators does not deviate from prior research or guidance; therefore, we believe these indicators provide a helpful nursing management perspective for minimizing physical restraints in practice. However, we did not practice test in feasibility and usability in practice for the number of indicators. We have added to the limitation on the feasibility and usability of the number of indicators in the revised manuscript. Please see page 30, lines 239–2341 and pages 31, 249–255.

Reviewer 2

1. When the population framework is known it is mandatory to use the probability sampling technique, and you need to state how you select the sample size.

Author Response: We appreciate your comments. We have explained our choice of the sampling technique in the revised manuscript. Please see pages 7, lines 118–122.

2. Another important an ethical issue the researcher have to consider is using AI in writing scientific paper also special consideration to the references, old ref should not be used.

Author Response: Thank you for your comments. We have not used any AI-assisted technologies in writing this article. Please see pages 9, lines 179. Further, as suggested by the reviewer and the editor, we have replaced some of the references that were published before 2013.

---

## [Decision Letter · Decision Letter 1]

26 Jun 2024

Development of management indicators of nursing for minimizing physical restraints focused on older adult patients hospitalized in acute care settings: a Delphi consensus study

PONE-D-23-31249R1

Dear Dr. Minamizaki,

We’re pleased to inform you that your manuscript has been judged scientifically suitable for publication and will be formally accepted for publication once it meets all outstanding technical requirements.

Kind regards,

Alejandro Botero Carvajal, MD

Academic Editor

PLOS ONE

Additional Editor Comments (optional):

Reviewers' comments:

Reviewer's Responses to Questions

**Comments to the Author**

1. If the authors have adequately addressed your comments raised in a previous round of review and you feel that this manuscript is now acceptable for publication, you may indicate that here to bypass the “Comments to the Author” section, enter your conflict of interest statement in the “Confidential to Editor” section, and submit your "Accept" recommendation.

Reviewer #1: All comments have been addressed

Reviewer #2: All comments have been addressed

2. Is the manuscript technically sound, and do the data support the conclusions?

Reviewer #1: Yes

Reviewer #2: Yes

3. Has the statistical analysis been performed appropriately and rigorously? 

Reviewer #1: Yes

Reviewer #2: Yes

4. Have the authors made all data underlying the findings in their manuscript fully available?

Reviewer #1: Yes

Reviewer #2: Yes

5. Is the manuscript presented in an intelligible fashion and written in standard English?

Reviewer #1: Yes

Reviewer #2: Yes

6. Review Comments to the Author

Reviewer #1: (No Response)

Reviewer #2: (No Response)

7. PLOS authors have the option to publish the peer review history of their article (what does this mean?). If published, this will include your full peer review and any attached files.

Reviewer #1: No

Reviewer #2: **Yes: **Dr. Nahla Elradhi Abdulrahman

---

## [Editor Report · Acceptance letter]

1 Jul 2024

PONE-D-23-31249R1 

PLOS ONE

Dear Dr. Minamizaki, 

I'm pleased to inform you that your manuscript has been deemed suitable for publication in PLOS ONE. Congratulations! Your manuscript is now being handed over to our production team.

Kind regards, 

on behalf of

Dr. Alejandro Botero Carvajal 

Academic Editor

PLOS ONE